# Inflammatory Cytokine-Induced Muscle Atrophy and Weakness Can Be Ameliorated by an Inhibition of TGF-β-Activated Kinase-1

**DOI:** 10.3390/ijms25115715

**Published:** 2024-05-24

**Authors:** Mai Kanai, Byambasuren Ganbaatar, Itsuro Endo, Yukiyo Ohnishi, Jumpei Teramachi, Hirofumi Tenshin, Yoshiki Higa, Masahiro Hiasa, Yukari Mitsui, Tomoyo Hara, Shiho Masuda, Hiroki Yamagami, Yuki Yamaguchi, Ken-ichi Aihara, Mayu Sebe, Rie Tsutsumi, Hiroshi Sakaue, Toshio Matsumoto, Masahiro Abe

**Affiliations:** 1Department of Bioregulatory Sciences, Tokushima University Graduate School of Biomedical Sciences, Tokushima 770-8503, Japan; kanai.mai@tokushima-u.ac.jp; 2Department of Hematology, Endocrinology and Metabolism, Tokushima University Graduate School of Biomedical Sciences, Tokushima 770-8503, Japan; byambasuren.ganbaatar11@gmail.com (B.G.); yky@tokushima-u.ac.jp (Y.O.); jumptera@okayama-u.ac.jp (J.T.); mitsui.yukari@tokushima-u.ac.jp (Y.M.); hara.tomoyo@tokushima-u.ac.jp (T.H.); m-shiho@tokushima-u.ac.jp (S.M.); yamagami.hiroki@tokushima-u.ac.jp (H.Y.); yamaguchi.yuuki@tokushima-u.ac.jp (Y.Y.); masabe@tokushima-u.ac.jp (M.A.); 3Department of Oral Function and Anatomy, Okayama University Graduate School of Medicine, Dentistry, and Pharmaceutical Sciences, Okayama 700-8570, Japan; 4Department of Orthodontics and Dentofacial Orthopedics, Tokushima University Graduate School of Biomedical Sciences, Tokushima 770-8503, Japan; tenhiro@tokushima-u.ac.jp (H.T.); c301951005@tokushima-u.ac.jp (Y.H.); mhiasa@tokushima-u.ac.jp (M.H.); 5Department of Community Medicine and Medical Science, Tokushima University Graduate School of Biomedical Sciences, Tokushima 770-8503, Japan; ihara@tokushima-u.ac.jp; 6Department of Clinical Nutrition, Faculty of Health Science and Technology, Kawasaki University of Medical Welfare, Okayama 700-8570, Japan; sebe@mw.kawasaki-m.ac.jp; 7Department of Nutrition and Metabolism, Tokushima University Graduate School of Biomedical Sciences, Tokushima 770-8503, Japan; mocoerie@live.com (R.T.); hsakaue@tokushima-u.ac.jp (H.S.); 8Fujii Memorial Institute of Medical Sciences, Tokushima University, Tokushima 770-8503, Japan; toshio.matsumoto@tokushima-u.ac.jp

**Keywords:** TAK1 inhibitor, myostatin, MyoD1, TNF-α, IL-1β

## Abstract

Chronic inflammation causes muscle wasting. Because most inflammatory cytokine signals are mediated via TGF-β-activated kinase-1 (TAK1) activation, inflammatory cytokine-induced muscle wasting may be ameliorated by the inhibition of TAK1 activity. The present study was undertaken to clarify whether TAK1 inhibition can ameliorate inflammation-induced muscle wasting. SKG/Jcl mice as an autoimmune arthritis animal model were treated with a small amount of mannan as an adjuvant to enhance the production of TNF-α and IL-1β. The increase in these inflammatory cytokines caused a reduction in muscle mass and strength along with an induction of arthritis in SKG/Jcl mice. Those changes in muscle fibers were mediated via the phosphorylation of TAK1, which activated the downstream signaling cascade via NF-κB, p38 MAPK, and ERK pathways, resulting in an increase in myostatin expression. Myostatin then reduced the expression of muscle proteins not only via a reduction in MyoD1 expression but also via an enhancement of Atrogin-1 and Murf1 expression. TAK1 inhibitor, LL-Z1640-2, prevented all the cytokine-induced changes in muscle wasting. Thus, TAK1 inhibition can be a new therapeutic target of not only joint destruction but also muscle wasting induced by inflammatory cytokines.

## 1. Introduction

Chronic inflammation due to the enhanced production of inflammatory cytokines causes muscle wasting with a reduction in muscle mass and power. Muscle wasting may increase the risk of falls, with a reduction in the activities of daily life [1]. Actions of inflammatory cytokines, IL-1β and TNF-α, as well as TGF-β, are mediated via phosphorylation and the activation of TGF-β-activated kinase-1 (TAK1) [2]. Because most of the inflammatory cytokine signals are mediated via TAK1 activation, inflammatory cytokine-induced muscle wasting may be ameliorated by the inhibition of TAK1 activity.

SKG mice, harboring the mutation of *ZAP-70* gene on the BALB/c background, spontaneously develop CD4^+^ T-cell-mediated autoimmune arthritis resembling human rheumatoid arthritis [3]. A spontaneous but slow and unpredictable development of arthritis is seen in SKG mice under conventional conditions, while mice under specific pathogen-free (SPF) conditions remain healthy. However, arthritis can be reproducibly induced by injecting mannan as an adjuvant under SPF conditions [4]. Because arthritis is mild and does not cause severe arthralgia, these mice can move freely and do not develop immobilization-induced muscle atrophy. Therefore, although SKG/Jcl mice are an arthritis model, mild arthritis does not cause immobilization-induced muscle atrophy, and most changes in the muscle can be regarded as inflammatory cytokine-induced muscle atrophy.

Mannan is a water-soluble and biodegradable polysaccharide produced by plants and fungi that acts as an adjuvant to enhance immune response [5]. Although SKG mice actively produce IL-6 as well as IL-1β and TNF-α, the signaling pathway downstream of gp130, a signal transducing subunit of the IL-6 receptor complex, involves JAK-STAT activation but does not involve TAK1 [6]. In addition, IL-6 expression in satellite cells and growing myofibers is enhanced by loading to the muscle, and IL-6 is a pro-myogenic cytokine. IL-6 is reported to positively regulate satellite cell activation and myoblast proliferation, as well as to enhance myogenic differentiation, myoblast migration, and fusion [7]. Thus, TAK1 inhibition does not affect IL-6 action, and IL-6 does not appear to cause muscle wasting.

We have previously demonstrated that TAK1 inhibition by LL-Z1640-2 (LLZ) reduced joint inflammation and bone destruction by inhibiting NLRP3 inflammasome and TNF-α expression in collagen-induced arthritis, a mouse model of rheumatoid arthritis [8]. Because rheumatoid arthritis also causes muscle wasting, our previous results suggest that TAK1 inhibition can be a therapeutic target of not only joint destruction but also muscle wasting by rheumatoid arthritis.

The present study was undertaken to clarify whether TAK1 inhibition can ameliorate inflammation-induced muscle wasting and can become a new therapeutic approach against the inflammatory cytokine-induced reduction in muscle mass and strength. In order to answer these questions, we treated SKG/Jcl mice with a small amount of mannan as an adjuvant and used LLZ to inhibit TAK1 activity, as reported previously [8]. We found that TAK1 inhibition successfully ameliorated inflammation-induced muscle wasting.

It is also important to clarify the signaling pathway downstream TAK1 activation that reduces muscle mass and strength. The present study demonstrated that TAK1 activation enhanced ERK, p38 MAPK, and NF-κB signaling, which enhanced the myostatin expression to suppress myoblast differentiation via suppressing MyoD1 expression and upregulated muscle protein degradation via enhancing Atrogin-1 and Murf1 expression.

## 2. Results

### 2.1. Inflammatory Cytokine-Induced Myopathy Is Ameliorated by a TAK1 Inhibitor LLZ In Vivo

To investigate whether a TAK1 inhibitor, LLZ, can affect inflammatory cytokine-induced muscle atrophy and weakness, we used an inflammatory cytokine excess model in SKG/Jcl mice with mannan-induced arthritis [3,9]. One injection of 1 mg of mannan to 8-week-old SKG/Jcl mice induced an elevation of both serum TNF-α and IL-1β at 10 to 12 weeks of age, and LLZ did not affect the increase in those cytokine levels (Figure 1A). Treatment with mannan or LLZ did not affect food intake (Figure 1B), while temporary body weight loss was observed in mice treated with mannan (Figure 1C). Mannan-induced inflammation model mice exhibited a mild increase in arthritis score, while the administration of LLZ from 8 to 12 weeks suppressed the increase in arthritis score (Figure 1D). LLZ also ameliorated a reduction in hindlimb muscle volume (Figure 2A) and the forelimb muscle weakness of SKG/Jcl RA model mice (Figure 2B).

Skeletal muscles have two types of fibers, type I slow-twitch and type II fast-twitch fibers. The anterior tibialis muscle is composed of predominantly type II fiber, while soleus muscle has almost equal numbers of type I and II fibers. In order to examine which type of fiber was predominantly affected by cytokine-induced inflammation, the area and number in each type of fibers were examined. As shown in Figure 3, there was a reduction in fiber sizes in both type I slow-twitch and type II fast-twitch fibers in mannan-treated mice. Measurements of the myofiber area of type I slow-twitch and type II fast-twitch fibers (Figure 4A) and myofiber number in anterior tibialis and soleus muscles (Figure 4B) revealed that the mean reduction in types I and II fiber sizes in anterior tibialis muscle by mannan treatment was 10.5% and 10.4%, respectively, and that in soleus muscle was 11.3% and 10.9%, respectively. When those mice were co-administered with LLZ, the inflammatory cytokine-induced muscle fiber atrophy was prevented in both anterior tibialis and soleus muscles. A number of type I and II fibers were not affected by any of the treatments (Figure 4B). These results demonstrate that mannan-induced inflammation reduces skeletal muscle volume and strength, and that the degree of reduction in muscle size is similar between type I and II fibers. Because the numbers of type I and II fibers were not affected by cytokine-induced inflammation, this suggested that fiber type switching does not occur in mice with systemic inflammation-induced myopathy. It is of note that all the above changes induced by mannan treatment can be ameliorated by the simultaneous inhibition of TAK1 activity by LLZ.

### 2.2. Effects of TNF-α and IL-1β on Myostatin and MyoD1 Expression Were Reversed by LLZ In Vitro

To clarify the mechanism whereby TAK1 inhibition can ameliorate inflammatory cytokine-induced myopathy, we performed in vitro experiments using a myoblast-like cell line, C2C12, under a low serum myogenic differentiation medium. In order to determine the optimal concentration of LLZ, we examined the dose-dependent effect of LLZ on the mRNA expression of *myostatin*, a suppressor of myogenic differentiation, and *MyoD1*, an enhancer of myogenic differentiation, in the presence of TNF-α. The expression of myostatin tended to decrease with LLZ administration, and LLZ enhanced *MyoD1* expression in C2C12 cells. TNF-α markedly enhanced *Myostatin* expression and suppressed *MyoD1* expression. LLZ at 3 μM slightly affected the expression levels of *myostatin* and *MyoD1*, but at 6 μM or higher, LLZ stably counteracted the effect of TNF-α in C2C12 cells (Figure 5A,B). Based upon these results, 6 μM of LLZ was used to counteract the effect of inflammatory cytokines in the later experiments.

We next examined the effect of 6 μM of LLZ on the mRNA expression of *myostatin* and *MyoD1* in C2C12 cells treated with TNF-α and/or IL-1β (Figure 6A). *The myostatin* mRNA expression was enhanced by TNF-α or IL-1β, while simultaneous treatment with LLZ suppressed *myostatin* mRNA expression. The addition of IL-1β together with TNF-α markedly enhanced *myostatin* mRNA expression, and the addition of LLZ suppressed the expression of *myostatin* mRNA (Figure 6A). The expression of *MyoD1*, a stimulator of the synthesis of skeletal muscle structural proteins, was upregulated by treatment with LLZ and was downregulated in the presence of TNF-α and/or IL-1β. The addition of LLZ abrogated the suppression of *MyoD1* mRNA expression by those cytokines (Figure 6B). 

### 2.3. Reversal of the Effects of TNF-α and IL-1β on the Expression of Muscle-Specific Proteins, Muscle Atrophic Factors, and Myotube Size by LLZ In Vitro

The expression of *Myh1* gene coding for skeletal myosin heavy chain (MHC) IIx (Figure 7A), *Myh4* gene coding for MHC IIb (Figure 7B), and *Myh7* gene coding for a slow-twitch muscle protein, MHC β (Figure 7C) [10], was reduced by the addition of TNF-α and/or IL-1β, and LLZ treatment abrogated the effects of those inflammatory cytokines.

The expression of muscle atrophic factor genes, *Atrogin*-*1* and *Murf*-*1* [11], was also enhanced by TNF-α and/or IL-1β, while LLZ treatment reversed the enhanced expression of *Atrogin*-*1* and *Murf*-*1* (Figure 8A,B). These results demonstrate that TAK1 inhibition ameliorates inflammatory cytokine-induced myopathy via a suppression of the expression of a muscle atrophic factor, myostatin, and an enhancement of the expression of myogenic factor, MyoD1, with the increased expression of both fast-twitch and slow-twitch muscle proteins, and that TAK1 inhibition by LLZ also suppresses the degradation of muscle proteins via a reduction in the expression of E3 ubiquitin ligases, Atrogin-1, and Murf-1. 

These effects of TAK1 inhibition by LLZ under treatment with inflammatory cytokines on muscle differentiation and degradation were reflected to the width of myotubes. Myotube size was reduced by those cytokines, and LLZ treatment canceled all the inflammatory cytokine-induced atrophy of the myotube (Figure 8C). Again, the expression of muscle-specific proteins and muscle atrophic factor genes as well as myotube width was even larger than the controls in the presence of LLZ.

### 2.4. TNF-α and IL-1β Enhance and LLZ Abrogates TAK1 Signaling In Vitro

Finally, the effect of TAK1 inhibition by LLZ on signaling pathway downstream cytokine stimulation was examined in vitro using C2C12 cells. In the presence of TNF-α or IL-1β, TAK1 was phosphorylated, and LLZ potently suppressed the TNF-α- or IL-1β-induced phosphorylation of TAK1 in C2C12 cells (Figure 9A,B). We next examined the effect of LLZ on intracellular signaling downstream TAK1 activation in C2C12 cells. TNF-α (Figure 9A) and IL-1β (Figure 9B) promptly induced the degradation of IκBα and the phosphorylation of p38 MAPK and ERK in serum-starved media. Treatment with LLZ abolished those cytokine-induced signaling cascade downstream TAK1 in C2C12 cells (Figure 9A,B). These results demonstrate that the effect of inflammatory cytokines on the reduction in muscle mass and power is mediated by the TAK1 signaling pathway via the activation of NF-κB, ERK, and p38 MAPK signaling. LLZ inhibits the phosphorylation of TAK1, thereby alleviating inflammatory responses through the suppression of downstream pathways such as NF-κB, ERK, and p38 MAPK signaling. Based upon these observations, TAK1 inhibition has the potential to serve as a specific therapeutic agent against inflammatory cytokine-induced muscle weakness and atrophy.

## 3. Discussion

The present study demonstrated that inflammatory cytokines, TNF-α and IL-1β, induced muscle weakness and atrophy in both type I and II muscle fibers and that the inhibition of TAK1 activity by LLZ prevented cytokine-induced changes in muscle fibers in SKG/Jcl mice. Those changes in muscles were mediated by an increase in myostatin expression and a reduction in *MyoD1* expression, causing a reduction in the expression of both fast-twitch and slow-twitch muscle proteins. There was also an enhancement of the degradation of muscle proteins by inflammatory cytokines via the enhanced expression of *Atrogin*-*1* and *Murf*-*1*. All these changes in muscle fibers were prevented by the inhibition of TAK1 activity by LLZ via the inhibition of TAK1 phosphorylation, resulting in the suppression of NF-κB, ERK, and p38 MAPK signaling.

Under physiological conditions, TAK1 is known to suppress the production of reactive oxygen species and oxidative stress in skeletal muscles [12,13] and is required to maintain satellite cells, muscle regeneration, and muscle function [14]. In contrast, TAK1 plays a pivotal role in disorders induced by inflammatory cytokines, including IL-1β and TNF-α via the stimulation of both MAPK and NF-κB signaling [2,15]. The present results demonstrate that the administration of LLZ can prevent the development of muscle atrophy and weakness under enhanced IL-1β and TNF-α expression induced by mannan in SKG/Jcl mice. These results are consistent with the assumption that this cytokine-induced muscle wasting can be inhibited by a blockade of TAK1 activity by LLZ.

The present study demonstrated that inflammatory cytokine signals enhance myostatin expression and suppress MyoD1 expression via TAK1 activation. It has been reported that myostatin inhibits myoblast differentiation and muscle protein synthesis by downregulating *MyoD1* and *myogenin* expression [16,17]. Myostatin is also shown to enhance the expression of *Atrogin*-*1* and *Murf1* [18]. The present results are consistent with those previous observations and support the notion that the enhancement of myostatin expression by inflammatory cytokines causes a suppression of myoblast differentiation via suppressing MyoD1 expression on the one hand and an enhancement of muscle protein degradation via enhancing Atrogin-1 and Murf1 expression on the other hand.

The expression of both TNF-α and IL-6 was enhanced in SKG/Jcl mice [3], and LLZ reduced the expression of both of them [8]. However, muscle wasting in those mice was mostly caused by enhanced myostatin expression by TNF-α but not by IL-6. In addition, the IL-6 signaling pathway is mainly JAK-STAT signaling, which does not involve TAK1. Therefore, although the expression of both TNF-α and IL-6 was reduced by LLZ, LLZ inhibited the TAK1-mediated effect of TNF-α but did not inhibit IL-6 action. Thus, even though the IL-6 expression was reduced by LLZ, that was not expected to influence negatively the effect of LLZ in muscles.

We have previously demonstrated that TAK1 inhibition by LLZ reduced joint inflammation and bone destruction by inhibiting NLRP3 inflammasome and TNF-α expression in collagen-induced arthritis, a mouse model of rheumatoid arthritis [8]. The present study adds another effect of TAK1 inhibition on muscle weakness and atrophy induced by inflammatory cytokines. Because rheumatoid arthritis also causes muscle wasting, the present observations, along with those previous results, suggest that TAK1 inhibition can be a therapeutic target for the treatment of rheumatoid arthritis by preventing both joint destruction and muscle wasting.

There are some limitations to this study. First, when LLZ was added along with TNF-α and/or IL-1β, an improvement in inflammatory cytokine-induced changes surpassing the control was observed in many myogenic and muscle atrophic factors or muscle-specific protein expression in C2C12 cells. It may be possible to explain those observations. Because TAK1 is also involved in the signaling cascade downstream myostatin [19], the inhibition of TAK1 may further suppress the action of myostatin on myogenic and muscle atrophic factors. However, the exact mechanism of this enhancement remains unclear. Second, we only assessed one compound of the TAK1 inhibitor. In the future, it will be necessary to consider using inhibitors other than LLZ or exploring means to suppress TAK1 expression. Third, in in vitro studies, the expression of myogenic genes, skeletal muscle-specific proteins, and muscle atrophy factors was assessed by only mRNA expression, and the evaluation of protein expression has not been performed. Fourth, the long-term effects of LLZ administration in vivo have not been investigated. These problems have to be clarified in future studies.

In conclusion, the present study demonstrated that TNF-α and IL-1β induced muscle weakness and atrophy in both type I and II muscle fibers and that the inhibition of TAK1 activity by LLZ prevented cytokine-induced changes in muscle fibers in SKG/Jcl mice. Those changes in muscle fibers were mediated via the phosphorylation of TAK1 that activated the downstream signaling cascade via the NF-κB, p38 MAPK, and ERK pathways, resulting in an increase in myostatin expression. Myostatin then reduced the expression of muscle proteins via a reduction in MyoD1 expression. Myostatin also caused the degradation of muscle proteins via enhancing Atrogin-1 and Murf1 expression. All these changes in muscle fibers were prevented by the inhibition of TAK1 activity by LLZ via the inhibition of TAK1 phosphorylation.

## 4. Material and Methods

### 4.1. Reagents

The following reagents were purchased from the indicated manufactures: mouse monoclonal antibodies against TNF-α and IL-1β, rabbit polyclonal antibodies against TAK1, IκBα, rabbit monoclonal antibodies against phosphorylated p38 MAPK, p38 MAPK, phosphorylate ERK, ERK, β-actin, horseradish peroxidase (HRP) anti-rabbit IgG, and anti-mouse IgG from Cell Signaling Technology (Beverly, MA, USA); antibodies against myosin heavy chain and goat IgG-FITC from Santa Cruz (Dallas, TX, USA); rabbit polyclonal anti-phosphorylated TAK1 from Cusabio (Houston, TX, USA); rabbit polyclonal antibodies against albumin from Invitrogen (Waltham, MA, USA); TAK1 inhibitor, LL-Z1640-2 (LLZ), from BioAustralis (Smithfield, NSW, Australia); recombinant human IL-1β from Wako (Osaka, Japan); recombinant human TNF-α from R&D Systems (Minneapolis, MN, USA); mannan from Sigma-Aldrich (St. Louis, MO, USA).

### 4.2. Animal Experiments

All animal experiments were performed in accordance with the guidelines of the Animal Research Committee, Tokushima University (approval number: T2020-75, date: 1 October 2020), and of the National Institutes of Health Guide for the Care and Use of Laboratory Animals. This study was approved by the Genetic Modification Experiment Safety Management Committee of Tokushima University (approval number: 2022-137, date: 9 April 2022). Eight-week-old female SKG/Jcl mice [3,20] were obtained from CLEA Japan (Tokyo, Japan) and were studied from 8 to 12 weeks of age. All mice were housed in standard filtered boxes under the specific pathogen-free conditions, 12 h light/dark cycle (0700-1900 on, 1900-0700 off) in 22–25 °C at 60 ± 5% humidity with free access to water and food. They were fed normal chow. The food intake was assessed by measuring the difference between feed offered and feed residue after 24 h of feed offering. In a previous report, in order to induce inflammation in SKG/Jcl mice, as high as 20 mg of mannan was given to SKG/Jcl mice [4]. However, a high amount of mannan treatment caused overt arthritis, which made it difficult to evaluate the muscle-specific inflammatory response induced by mannan. Therefore, we titrated the dose of mannan and found that as low as 1 mg of mannan was able to cause muscle wasting with only minor arthritic changes.

SKG/Jcl female mice were randomly divided into four experimental groups. Group 1 was the control group injected with vehicle alone, and group 2 was intraperitoneally injected with 1 mg/kg TAK1 inhibitor, LLZ, 3 times a week from 8 to 12 weeks of age. Group 3 was intraperitoneally injected with 1 mg of mannan at 8 weeks of age, and group 4 was injected with 1 mg of mannan at 8 weeks along with 1 mg/kg LLZ 3 times a week from 8 to 12 weeks of age. Each group consisted of 12 mice.

At the end of the experiment, animals were euthanized using isoflurane; their tibialis anterior and soleus muscles were dissected, removed, rapidly frozen in liquid nitrogen, and stored at −80 °C until processing.

### 4.3. Western Blot Analysis of Serum TNF-α and IL-1β

To measure serum TNF-α and IL-1β, mouse serum was precipitated with cold acetone overnight and dissolved using lysis buffer (Thermo Fisher Scientific, Lafayette, CO, USA) supplemented with 1 mM phenylmethylsulfonyl fluoride and a protease inhibitor cocktail solution (Sigma, St. Louis, MO, USA). The relative amount of these proteins was detected by Western blotting, as described previously [8].

### 4.4. Arthritis Score

To assess the severity of mannan-induced arthritis, the arthritis score was monitored, as previously described [21]; score 0, no joint swelling; 0.1, swelling of one finger joint; 0.5, mild swelling of one wrist or ankle; and 1.0, severe swelling of one wrist or ankle. The total scores for all fingers of forepaws and hind paws, wrists, and ankles were calculated in each mouse.

### 4.5. Micro-Computed Tomography (μCT) Analysis

Before and during the CT scan, mice were anesthetized by the inhalation of isoflurane (Abbott, Tokyo, Japan). Mice were placed in an abdominal position in a 48 mm wide specimen holder with 96 μm pixel resolution. Their hind limbs were extended to keep the femur and spine at a right angle, and then they were scanned from the proximal end to the distal end of the tibia to measure the muscle volume of their lower limbs. The scanning μCT device was LaTheta LCT-200 (Hitachi-Aloka, Tokyo, Japan). The micro-CT analysis of the lower limb muscle volume was performed at 12 weeks just before sacrifice.

### 4.6. Measurement of Grip Strength

The Animal Grip Strength System (MK-380CM/F, Muromachi Kikai, Tokyo, Japan) was used and grip strength was measured at the age of 8 and 12 weeks. Mice were placed on a small mesh and grasped the Animal Grip Strength System apparatus with their forepaws. The tails of the mice were then slowly pulled away from the apparatus until they released the grip apparatus. The apparatus automatically measured the maximum grip force of the mice at the time of release.

### 4.7. Muscle Histology

After μCT scanning, the skeletal muscles taken from the lower hind limbs were further processed for histology. Paraffin sections of 5 μm thickness of the anterior tibialis and soleus were prepared and stained with NADH-tetrazolium reductase (NADH-TR) staining to assess the fiber area and percentage of type I and II fibers. Staining was performed as described previously [22]. The fiber area and number were determined through blinded analyses with the Image J software ver 1.53j (National Institute of Health, Bethesda, MD, USA) of randomly captured 30 or 100 fibers per mouse from 3 mice in each group.

### 4.8. Cell Culture

Mouse C2C12 myoblasts (Riken Cell Bank, Tsukuba, Japan) were cultured in high glucose Dulbecco’s Modified Eagle’s Medium (DMEM) supplemented with 10% fetal bovine serum, 100 U/mL penicillin, and 100 μg/mL streptomycin and maintained at 37℃ in a humidified atmosphere of 5% CO_2_ in room air. Differentiation to myotubes was induced by shifting confluent cultures to DMEM supplemented with 2% heat-inactivated horse serum. The medium was changed every 2 days, and most of the cells were fused to form myotubes within 5 days [23]. Differentiated myotubes were treated with 100 ng/mL TNF-α and 0 to 15 μM LLZ for examining the dose dependency of LLZ, with 100 ng/mL TNF-α and/or 1 ng/mL IL-1β with or without 6 μM of LLZ for examining the treatment effect of LLZ for 48 h.

### 4.9. Immunofluorescence Microscopy and Myotube Analysis

Immunofluorescence microscopy analyses were carried out as described previously [24]. Antibodies against myosin heavy chain were diluted at 1:400, and secondary antibody anti-goat IgG-FITC was diluted at 1:500. The stained myotube cells were analyzed under a fluorescence microscope (BZ-X800; Keyence, Osaka, Japan). The myotube width was quantified through a blinded analysis using the Image J software (NIH, USA). The average myotube width was evaluated as the mean of five approximately equidistant measurements taken along the length of the myotube. For each treatment, 10 fields of view were chosen randomly, and 10 myotubes were measured in each field. 

### 4.10. Real-Time PCR

Total RNA was extracted from myotubes with TRIzol reagent (Invitrogen, Carlsbad, CA, USA) according to the manufacture’s protocol. The extracted RNA was subjected to qPCR analysis as described previously [25].

Sequences of primers are listed below:mouse *Myostatin* F: GAGCCCAGGCACTGGTATTT andR: AGGGATTCAGCCCATCTTCTCmouse *Myod1* F: TGCTCTGATGGCATGATGGATT andR: TGGAGATGCGCTCCACTATGmouse *Myh1* F: GGACCCACGGTCGAAGTTG andR: CCCGAAAACGGCCATCTmouse *Myh4* F: CAATCAGGAACCTTCGGAACAC andR: GTCCTGGCCTCTGAGAGCATmouse *Myh7* F: AGTCCCAGGTCAACAAGCTG andR: TTCCACCTAAAGGGCTGTTGmouse *Atogin1* F: TCTCAGAGAGGCAGATTCGC andR: TGAGGGGAAAGTGAGACGGAmouse *Murf1* F: GGGCTACCTTCCTCTCAAGTG andR: GAGCGTGTCTCACTCATCTCC

### 4.11. Western Blot Analysis of C2C12 Cells

C2C12 myotubes were lysed with ice-cold cell lysis buffer (Thermo Fisher Scientific, Lafayette, CO, USA). Western blot analysis of phosphorylated TAK1, IκBα, phosphorylated p38 MAPK, and phosphorylated ERK was performed as described previously [8].

### 4.12. Statistics

Data on body weight, lower muscle volume, grasping power, fiber area, fiber number, and width of myotubes are presented as means ± SE, and the arthritis score and mRNA expression of *myostatin*, *MyoD1*, *Myh1*, *Myh4*, *Myh7*, *Atrogin*-*1*, and *Murf*-*1* are presented as means ± SD. Statistical differences are analyzed by an ANOVA test followed with the appropriate post hoc test for multiple comparisons, using statistical software JMP 14.2 for Macintosh (SAS Institute Inc., Tokyo, Japan), and are described in each figure legend. Differences are considered statistically significant at a *p* value of 0.05.

## Figures and Tables

**Figure 1 ijms-25-05715-f001:**
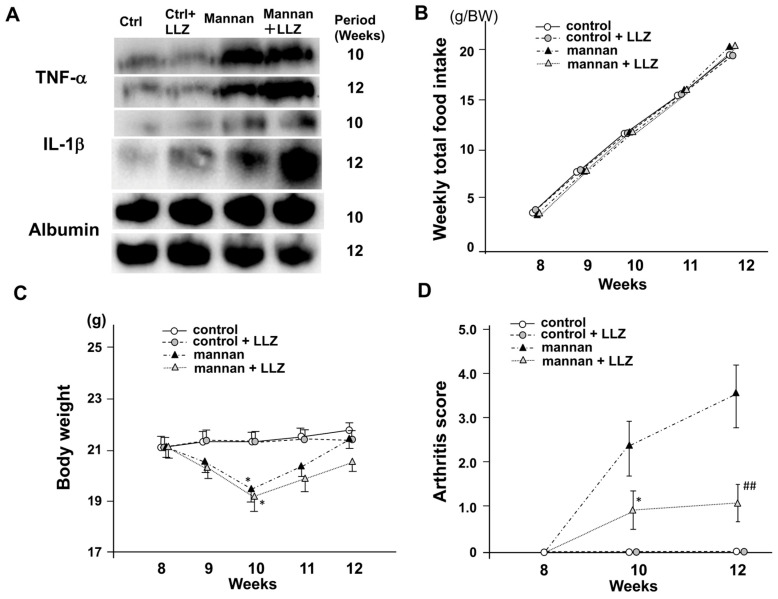
LLZ alleviates inflammation and arthritis in SKG/jcl mice with mannan-induced inflammatory cytokine excess. (**A**) Western blot analysis of serum concentration of TNF-α and IL-1β. Serum levels were increased in SKG/Jcl mice treated with mannan. (**B**) Accumulated weekly food intakes are expressed as g/g body weight (BW) (n = 4 for each group). (**C**) Weekly change in body weight from 8 to 12 weeks of age (n = 12 for each group). Data are means ± SE for 12 mice, * *p* < 0.05 vs. control or control group with LLZ. (**D**) Change in arthritis score measured at all 4 paws in each mouse at 8, 10, and 12 weeks. The sum of scores for 4 paws in each mouse is presented. Data are means ± SD for 4 mice per each group, * *p* < 0.05, ## *p* < 0.001 vs. mannan group.

**Figure 2 ijms-25-05715-f002:**
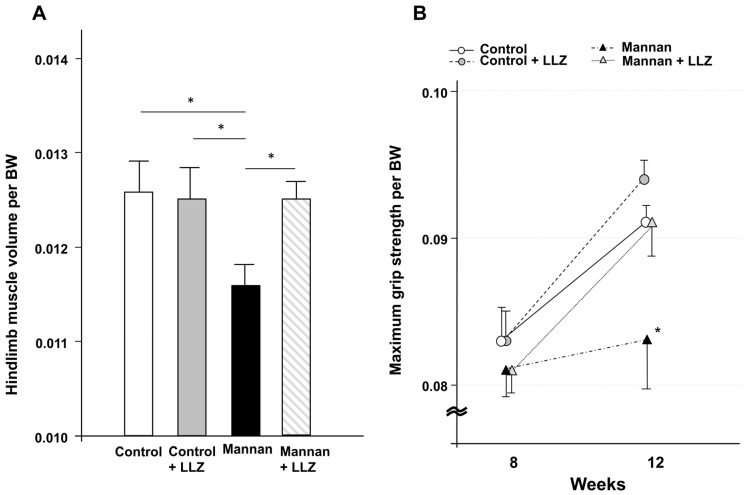
LLZ ameliorates inflammatory cytokine-induced muscle wasting. (**A**) Hindlimb muscle volume corrected by body weight. Data are expressed as means ± SE for 10 mice per each group, * *p* < 0.05. (**B**) Maximum grip strength corrected by body weight. Five consecutive measurements were recorded per mouse, and the average of the three best scores for each mouse is presented. Data are means ± SE for 12 mice per each group, * *p* < 0.05 vs. mannan + LLZ group.

**Figure 3 ijms-25-05715-f003:**
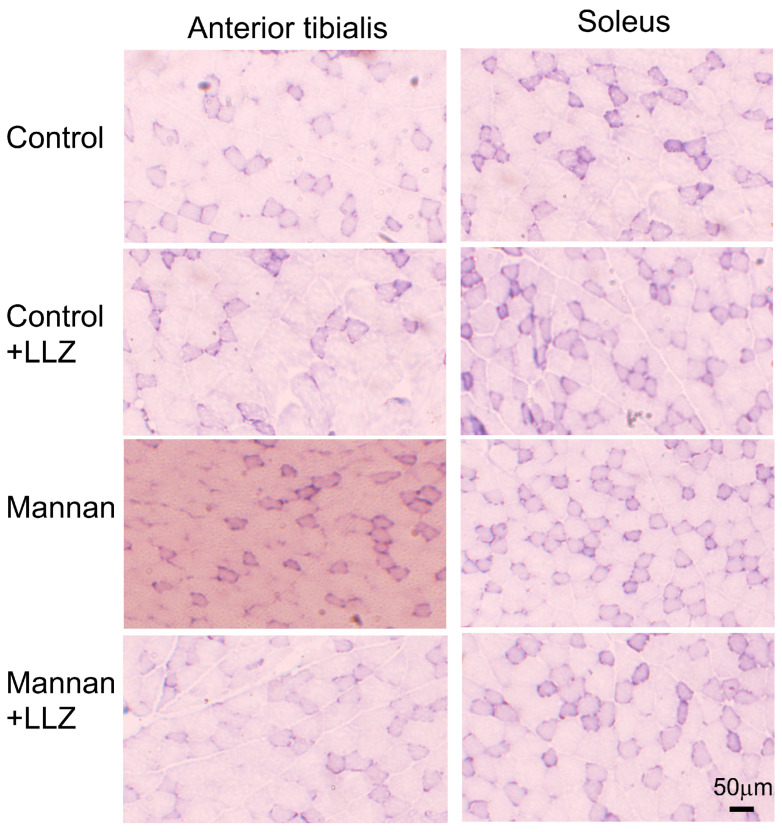
Histological images of cross-sections of anterior tibialis and soleus muscles stained by NADH-TR. Granular muscle fibers represent type I slow-twitch fibers, and lighter staining fibers represent type II fast-twitch fibers. Scale bar = 50 μm.

**Figure 4 ijms-25-05715-f004:**
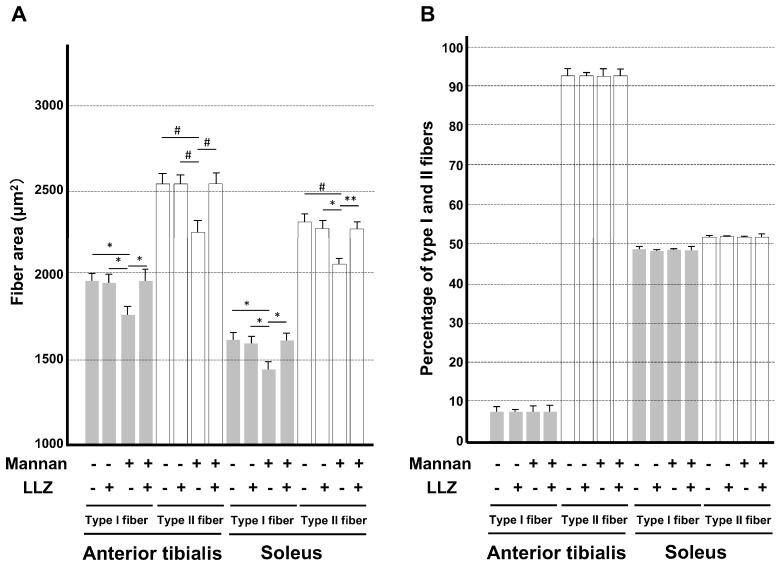
LLZ ameliorates inflammatory cytokine-induced muscle atrophy. (**A**) Average myofiber area of type I slow-twitch and type II fast-twitch fibers and (**B**) average myofiber number in anterior tibialis and soleus muscle. Data are expressed as means ± SE for 30 measurements per a mouse from 3 mice, * *p* < 0.05, ** *p* < 0.01, # *p* < 0.005. Gray bars represent type I slow-twitch fibers and white bars represent fast-twitch fibers.

**Figure 5 ijms-25-05715-f005:**
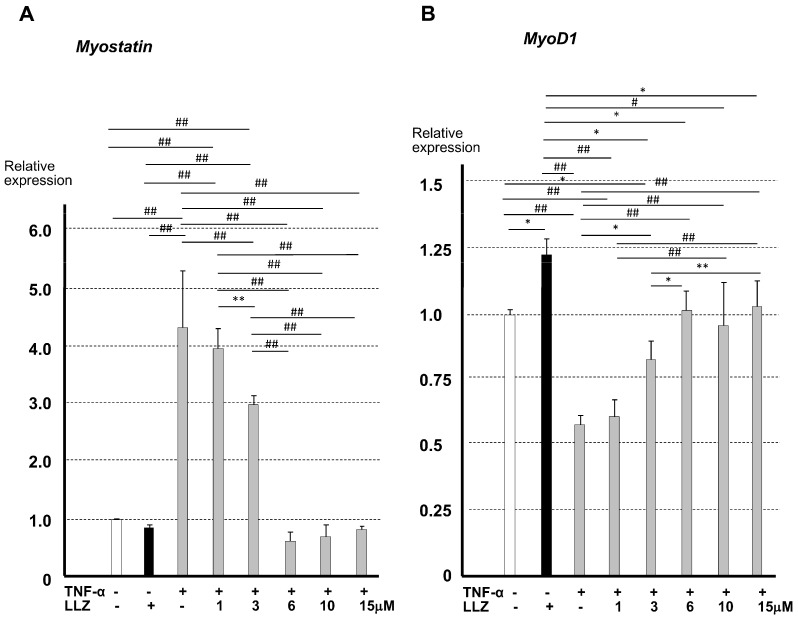
Enhancement of *myostatin* expression and suppression of *MyoD1* expression by TNF-α were reversed by LLZ at 6 μM or higher in C2C12 cells. Relative mRNA levels by qPCR of (**A**) *myostatin* and (**B**) *MyoD1* in C2C12 cells. Data are expressed as means ± SD for 5 measurements, * *p* < 0.05, ** *p* < 0.01, # *p* < 0.005, ## *p* < 0.001.

**Figure 6 ijms-25-05715-f006:**
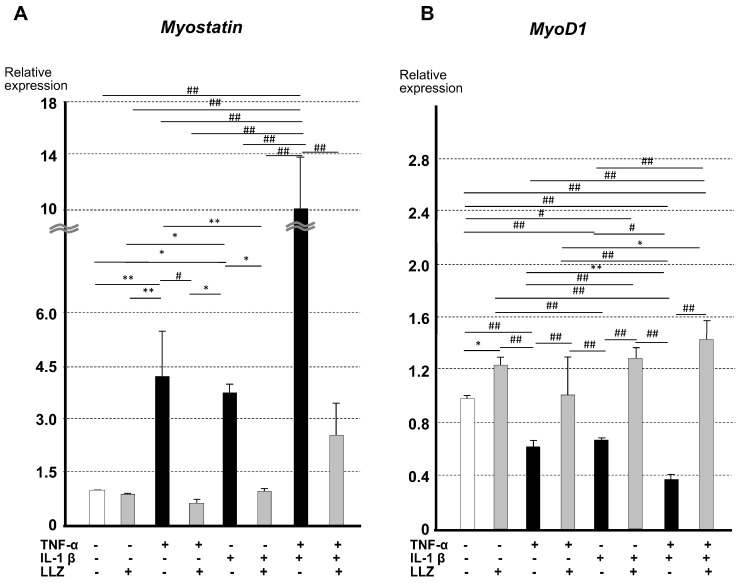
Enhancement of *myostatin* expression and suppression of *MyoD1* expression by TNF-α and IL-1β were reversed by LLZ in C2C12 cells. Relative mRNA expression levels by qPCR of (**A**) *myostatin* and (**B**) *MyoD1* in C2C12 cells. Data are expressed as means ± SD for 5 measurements, * *p* < 0.05, ** *p* < 0.01, # *p* < 0.005, ## *p* < 0.001.

**Figure 7 ijms-25-05715-f007:**
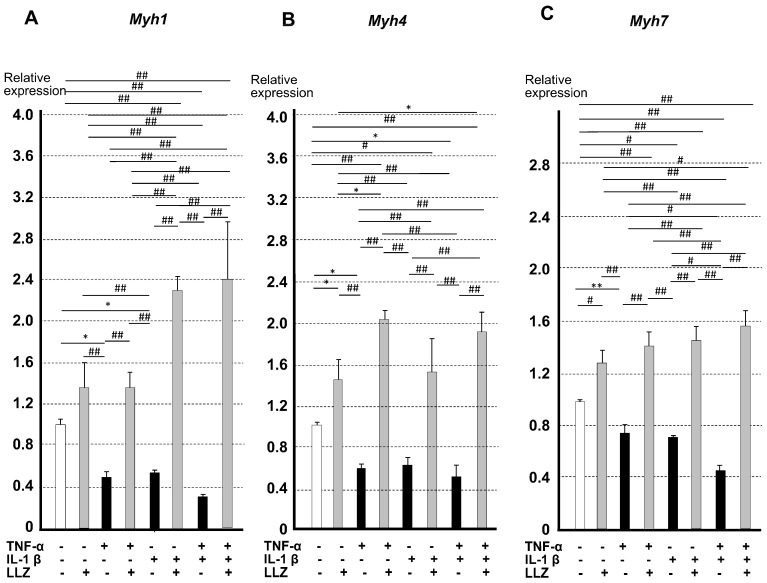
Effects of TNF-α and IL-1β on the mRNA expression of muscle-specific proteins were reversed by LLZ in C2C12 cells. Relative mRNA levels by qPCR of muscle-specific fast-twitch proteins, (**A**) *Myh1*, (**B**) *Myh4*, and a slow-twitch protein, (**C**) *Myh7*, in C2C12 cells. Data are expressed as means ± SD for 5 measurements, * *p* < 0.05, ** *p* < 0.01, # *p* < 0.005, ## *p* < 0.001.

**Figure 8 ijms-25-05715-f008:**
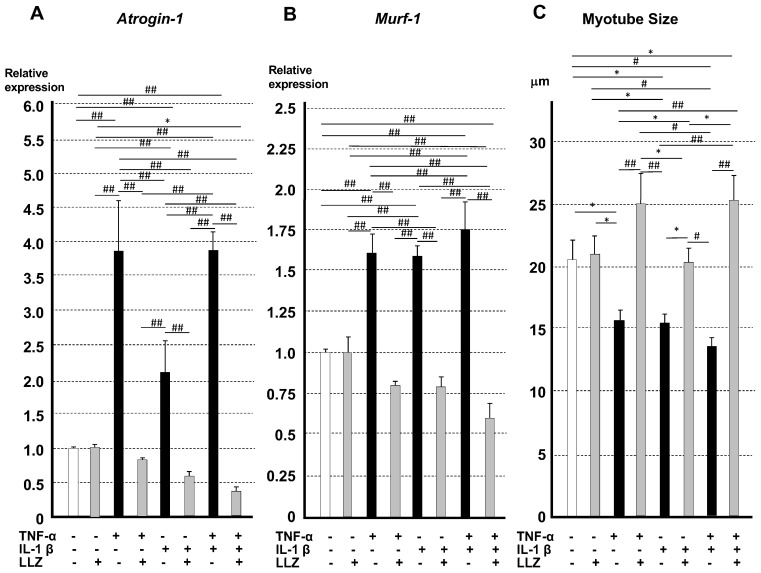
Effects of TNF-α and IL-1β on the mRNA expression of muscle atrophic factors and myotube size were reversed by LLZ in C2C12 cells. Relative mRNA levels by qPCR of (**A**) *Atrogin*-*1* and (**B**) *Murf*-*1* in C2C12 cells. Data are expressed as means ± SD for 5 experiments. (**C**) Size of myotube differentiated from C2C12 cells. Data are means ± SE for 10 measurements per each culture dish from 5 each treatment group, * *p* < 0.05, # *p* < 0.005, ## *p* < 0.001.

**Figure 9 ijms-25-05715-f009:**
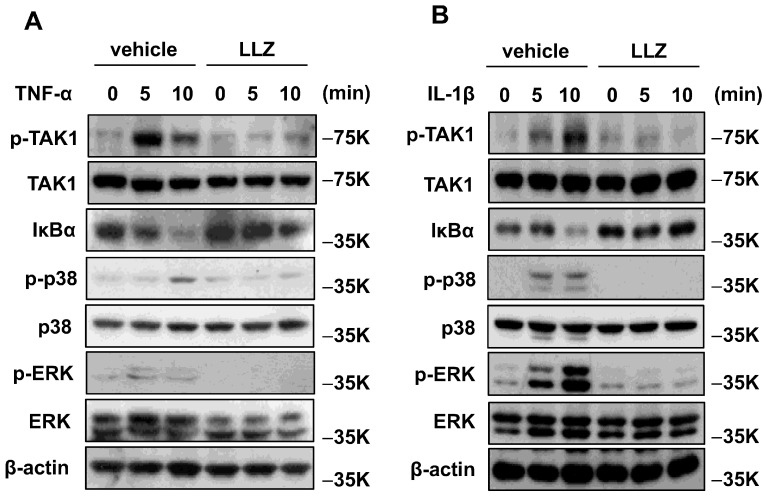
TNF-α and IL-1β enhanced TAK1 signaling, and LLZ abrogated TAK1 signaling in C2C12 cells. Western blot analysis of intracellular IκBα, phosphorylated TAK1, p38, and ERK in C2C12 cells treated with (**A**) TNF-α or (**B**) IL-1β in the presence or absence of LLZ.

## Data Availability

The data supporting the findings from this study are available within the manuscript.

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
