# Peer review of "Inflammatory Cytokine-Induced Muscle Atrophy and Weakness Can Be Ameliorated by an Inhibition of TGF-β-Activated Kinase-1"

_ijms, 2024, doi:10.3390/ijms25115715_

Round 1
Reviewer 1 Report
Comments and Suggestions for Authors
The main purpose of the work was to clarify whether TAK1 inhibition can ameliorate inflammation-induced muscle wasting.
I have studied the research in detail. I thank the authors for their efforts, this research is original and relevant. TAK1 inhibition may represent a viable and useful signals for evaluating and prevent muscle wasting. this research is original and relevant addressing the role of TAK1 inhibition as a useful signals for evaluating and prevent muscle wasting
The introduction is too short and fairly weak for readers. This section should be improved , I propose to expand the information on adverse outcomes of muscle wasting, such as risk of fall, with more directional literature citing papers such as DOI: 10.3390/ijms25031368
Insert the materials and methods chapter before the results
Discussion:
Conclusion are consistent but authors should add a limitations section.
Tables and figure are adequate for readers and for a scientific journal
Comments on the Quality of English Languageminor
Reviewer 2 Report
Comments and Suggestions for Authors
In the manuscript submitted to IJMS, the authors studied the amelioration of muscle atrophy by TAK-1 inhibition.
The manuscript is well-written and easy to follow.
However, the reviewer has some concerns to conclude that TAK-1 inhibition is beneficial for muscle atrophy.
The authors used the arthritis model, SKG/Jcl mice. The reviewer understands that the manuscript is the continuation study of the group. However, even if mannan level was reduced, this is the arthritis model. Therefore, it is not a good model to study muscle atrophy. The authors also discuss sarcopenia and inflammation/fibrosis. The current data does not support that TAK-1 inhibition is beneficial for sarcopenia.
In other papers, the authors stated that TAK1 inhibition with LLZ may become a novel treatment strategy to effectively alleviate inflammasome-mediated inflammation and RANKL-induced osteoclastic bone destruction in joints alongside its potent suppression of TNF-α and IL-6 production and proteinase-mediated pathological processes in RA. Do authors believe LLZ could also be useful for muscle wasting in addition to joint destruction? iL-6 could work to affect on muscle positively.
What is the specificity of LL-Z1640-2? The compound also has effects on the inhibition of MEK1/2 and MKK3/6. As far as the reviewer knows, LL-Z1640-2 is not clinically approved. The sentence in the Abstract on page 1, lines 41-42 is overstated.
On page 12, lines 287-289, the reviewer cannot agree with the discussion of the effect of small amounts of inflammatory cytokine in the serum.
Other comments
On page 2, line 48, cite a reference of a reduction in muscle mass and power if available.
Fig.1A, control such as GAPDH is needed.
Histological images of skeletal muscle should be shown in addition to graphs.
As the authors discuss, the protein level of myostatin and Myod1 is not studied.
In Fig.9, MyoD1 seems downstream of Myostatin. There are reports to show that myostatin is under the control of myogenic transcription factors, cAMP, and PPARgamma. Since only the time course of mRNA is shown in Fig.8, the authors cannot the order of expression of MSTN and MyoD in the current study. The mechanism of how NF-kB and AP-1 regulate Myostatin gene expression is not clear. Most of the molecules in Fig.9 are not studied or mentioned in the text.
Fig.8 legend, correct the last “LLZ…”.
Round 2
Reviewer 1 Report
Comments and Suggestions for Authors
Acceptable for publication if appropriate for the editor and other reviewers.
Comments on the Quality of English Language
Minor
Reviewer 2 Report
Comments and Suggestions for Authors
The manuscript has been revised properly, and is fine for proceed to acceptance.